# Caring for Your Child during COVID-19—Utilizing a Light-Touch Parenting Resource during Lockdown in Indonesia

**DOI:** 10.3390/ijerph19074046

**Published:** 2022-03-29

**Authors:** Aala El-Khani, Ali Yassine, Karin Haar, Narendra Narotama, Lucky Pramitasari, Melvi Rosilawati, Wadih Maalouf

**Affiliations:** 1Prevention, Treatment and Rehabilitation Section, Drug Prevention and Health Branch, Division of Operations, United Nations Office on Drugs and Crime (UNODC), Wagramer Strasse 5, A-1400 Vienna, Austria; ali.yassine@un.org (A.Y.); karin.haar@un.org (K.H.); wadih.maalouf@un.org (W.M.); 2Drug Demand Reduction Division, United Nations Office on Drugs and Crime (UNODC), Menara Thamrin Building 10th Floor, Central Jakarta, Jakarta 10250, Indonesia; narendra.narotama@un.org (N.N.); lucky.pramitasari@un.org (L.P.); 3Charitas Hospital Palembang, Central Jakarta, Jakarta 30129, Indonesia; melvi.rosilawati@gmail.com

**Keywords:** COVID-19, family skills, parenting, resilience, stress, lockdown

## Abstract

To tackle the spread of COVID-19 globally, countries around the world have responded by implementing measures such as lockdowns, social distance maintenance, temporary school closures, and remote working and learning. COVID-19 social isolation has been found to increase stress, and potentially have long term harmful effects on both mental and physical health. Stress and compromised parenting often place children at risk of violence and abuse. In parallel, times of hardship might also provide an opportunity to build stronger relationships with our children. The United Nations Office on Drugs and Crime (UNODC) joined many other agencies and international organizations in recognizing the threat the pandemic might have on individual and family wellbeing, and has thus availed a number of light-touch parenting resources. One such tool is the ‘Caring for your child in response to the COVID-19 lockdown’ booklet, developed to enhance parenting skills, and to build family harmony as challenged by the COVID-19 context. This short communication reflects on a feasibility study that took place in Indonesia during the implementation of this booklet with 30 parents in five cities. Thematic analysis identified challenges in parenting during COVID-19, as well as reported positive experiences of engaging in the parenting resource. The findings are discussed with regard to the usefulness of light-touch parenting information, adding to the context of the feasibility and global scalability of reaching families. The implications pave the way to the engagement and implication of more intensive parenting information interventions in high-stress contexts. Despite the challenge, there is promising news for families globally, as agencies and policy-makers begin to recognize the importance of supporting families with the appropriate skills to navigate extreme stress contexts with effective strategies.

The ongoing global COVID-19 pandemic has affected millions of people around the world in terms of infection, disease, and sequelae. To tackle this public health issue, and reduce the spread of SARS-CoV-2, countries around the world have implemented several measures, from lockdowns to social distance maintenance, temporary school closures, home-offices, and more. Further to the direct consequences of the pandemic and its economic impact, these measures carried a heavy burden on parenting practices, leaving caregivers overwhelmed and unsure how to best respond to their children’s anxieties, worries, behaviours, and needs. These challenges had an amplified differential effect on the millions of already vulnerable families worldwide, such as refugees, those who are internally displaced, or those living in low-resource contexts.

Social isolation has been found to increase stress, and may have harmful effects on both mental and physical health [1]. Stress and compromised parenting often place children at risk of violence, abuse, or neglect [2]. In parallel, times of hardship might also provide an opportunity to build stronger relationships with our children and adolescents [3]. There is a strong body of evidence now that indicates that supportive family environments may serve as protective factors for individuals experiencing stress, and may differentially influence abuse potential [4].

The United Nations Office on Drugs and Crime (UNODC) joined many other agencies and international organisations in recognising the threat the pandemic. UNODC availed a set of policy documents and tools to address several factors that have been affected differently by the pandemic. These included corruption, crime, criminal justice responses, violence, gender-based violence, cybercrime, as well as the prevention and treatment of drug use and disorders [5]. One of the core materials produced was to enhance family wellbeing through specific parenting under COVID-19 tools. Following a long experience of availing parenting materials for times of conflict, refuge, displacement, violence, and other forms of parenting under stress, UNODC developed light, open-access, caregiver resources in the form of leaflets and a booklet to support caregivers during the COVID-19 pandemic. These provide tips on developing relationships between family members, finding opportunities to spend one-to-one time with children, communication, encouraging good behaviour, and dealing with unwanted behaviour under such circumstances. UNODC also collaborated with its partner agencies, for example, under the INSPIRE initiative to end violence against children, to avail further brief infographics supporting other parenting skills [6]. All of these resources are based on components identified in evidence-based reviews, and are available to families online and in multiple languages.

The UNODC COVID-19 parenting tools are being utilized through UNODC’s ongoing family-skills-focused work globally. Such dissemination work is linked to a set of research studies underway monitoring the feasibility, as well as effectiveness, of such tools. The aim of this short communication is to describe the feasibility study that took place in Indonesia during the implementation of the ‘Caring for your child in response to the COVID-19 lockdown’ booklet [7].

Indonesia, like many countries globally, has implemented large-scale social restriction measures. Employed parents shifted to home working arrangements whilst managing children’s online schooling in parallel. This required family adjustments at home with a direct impact on the work–life balance. This often causes stress, and poses a negative impact on the mental wellbeing of parents and their children [8]. This carried a more significant toll on families whose income was affected or who even lost their jobs due to companies’ cut-down measures. Additionally, the heavier reliance on electronic tools and devices not only caused economic stress on parents, but was also a challenge for those not tech-savvy. Besides the direct cost of purchasing the materials, the cost of internet and phoneline credit and other necessary software or hardware needs placed an added economic burden.

The aim of the ‘Caring for your child in response to the COVID-19 lockdown’ booklet is to enhance caregivers’ parenting skills, and to develop family harmony as challenged by the COVID-19 context. Such skills work to improve the mental wellbeing of children and their families. It contains information to normalise the challenges caregivers may be facing, as well as their children, and what they can do to both help themselves and their children. Tips are provided on finding opportunities to spend one-to-one time with children, how to develop positive communication by listening and talking, how to encourage good behaviour by using clear instructions and praise, as well as dealing with unwanted behaviour with sensible consequences. Nine short video sketches were developed in the local language (Bahasa) in Indonesia to highlight key topics in the booklet in an engaging and clear way, and to overcome any challenges with low caregiver literacy. These topics included giving warmth and support, giving praise, spending time together, encouraging good behaviour, fighting and aggression, maintaining routines, encouraging play, fears and night disturbance, and relaxation techniques. The duration of each video varied from 3 to 6 min.

In March 2021, thirty parents (with at least one child aged 5–15) in five cities in Indonesia who had consented to take part after receiving a study information sheet, were provided with both printed and electronic copies of the ‘Caring for your child during COVID-19 lockdown’ booklet, as well as access to the associated video sketches. Instructions were given to caregivers to read the booklet first, and access the videos after 7 days. The five cities were identified as those that implemented Large-Scale Social Restrictions (PSBB), and were considered as ‘red zones’ for the spread of COVID-19. One week later, two researchers conducted semi-structured interviews with the parents, with each researcher leading 15 interviews each. Interviews explored the parenting experience of Indonesian families in caring for their children during the lockdown, as well as the usefulness and acceptability of the booklet and video sketches.

Results were interpreted using a thematic analysis, and were found to reflect two main themes. Firstly, the COVID-19 challenging parenting experience, and secondly, the experiences of parents with the booklet and video materials.

## 1. COVID-19 Challenging Parenting Experience

Regarding the parenting experience, parents described a great struggle adjusting to the changes that the lockdown brought to family life. They described school closures, remote working, a loss of income, and a lack of external childcare as being catalysts for increased levels of stress and worry. As one parent said: ‘*I became far more protective of my children and anxious because of the constant many fears we were faced with regarding catching COVID-19*’. Family conflict had increased due to lockdown restrictions being frustrating for children, with parents needing to ensure their children abide by them to keep safe. For example, children were no longer allowed to play outdoors, an activity which, prior to the lockdown, was a part of daily life. Parents felt they wanted ideas on how to entertain their children, and ideas to do family activities that were not based around electronic devices. One parent reflected: ‘*My children spent too much of their time on the phone to the point they lose track of time*’. Parents described their frustration at having to repeat the reasons they could not allow their children to leave their homes, while the parents themselves were also frustrated and wanted to go out. For many parents, they struggled with how to communicate clearly and calmy.

Parents also struggled with supporting their children through home-schooling. This was problematic for several reasons, such as the children not complying with participating in the online lessons, the parents themselves struggling to understand the school content when their children asked for help, and parents already being too busy working from home to be able to manage home-schooling. One parent added: ‘*Making time for them and work, as well as ensuring the days for the children are not boring is hard*’. This led to many parents describing feeling as though they did not have the skills to be the parent their children needed at this time, and this was very upsetting for them.

When asked what they believed were their biggest challenges with their children during lockdown, all parents mentioned an increase in their children’s misbehaviour, and not knowing how to deal with this in a calm and productive manner. They recognised they often lost their temper, and were regretful, but did not know how else to manage overwhelming feelings of stress and worry. One parent shared: ‘*Sometimes, we lose our temper too*’. They described being aware that they were generally angrier and more aggressive. They stated they were sometimes giving their children contradictory parenting messages when they were feeling burnt out. For example, they were allowing their children more time on electronic devices, as there was not much else they could do to entertain their children, but then reprimanding them with angry words and punishment when the children overused their devices. When asked how they felt their children were coping, parents mentioned their children were bored, stressed, angry, and worried. One parent described their child by saying: ‘*The children show mood swings when they felt bored. They have not understood the feeling of boredom, so vent by having tantrums*’. They felt a lack of control at not being able to help their children overcome these feelings.

Despite the numerous challenges, several parents were able to reflect on the positive impacts the lockdown had on their families. These were mainly related to the fact that they were able to now spend more time with their children at home. One parent said ‘*Well, on the positive side, previously we were not as close with the kids, now we’re growing closer, it’s a progress now to get close to them*’. They described how, prior to the lockdowns, they might only see their children in the evening after a long day of work, where they would be too tired to talk or play with their children. The lockdown allowed for much time to begin to enjoy family closeness, and to reconnect as a family. Parents described becoming better at recognizing and understanding their children’s emotions and needs.

## 2. Experiences of Parents with the Booklet and Video Materials

As for the experiences of parents with the booklet and video sketches, all parents provided positive feedback on the ‘Caring for your child during COVID-19 lockdown’ booklet and videos. They felt the booklet was simple to understand, and long enough to be engaging without being too long to overwhelm them. They mentioned that they found the inclusion of clear examples on the implementation of the activities and tips very useful. One parent added, ‘*The tips are very practical and straightforward to follow*’. For some parents, the booklet proved to be a self-reflection tool, giving them the opportunity to reflect on how they were parenting so far, and what they could do to improve. The booklet helped some parents adapt how they were parenting, and make a conscious effort to improve their interactions with their children. Another parent said, ‘*It taught me practical ways, which are simple and easy, to manage children’s behaviour with a sense of compassion. Sometimes we mean well, but we use the wrong way of communication which could be fatal for the kids. This book has been very helpful*’. Information on how their children were also likely overwhelmed and struggling to cope with the changes pushed parents to adopt a gentler empathetic approach with their children. ‘*I have never seen parenting tips focusing on children’s emotion. I realize now this makes sense. They are struggling the most, yet usually tips only focus on ourselves as parents*’.

Tips, such as those on listening and talking to children, were very welcomed. Parents described how, in just the few days after reading the booklet and beginning to implement the tips, they had already seen a difference in their family connections. One parent illustrated this by saying, ‘*We became closer and know each other due to more intensified communication with the kids. We learned to be more patient and compassionate*’. They described making a conscious effort to parent more calmly, and make time for one-to-one interactions with their children, and thus, attributed this to their children responding with an increase in the behaviour they wanted. Children began to open up about their worries and challenges, sharing their feelings with their parents. Parents particularly found the information on managing misbehaviour useful and clear, as it gave a stepped approach that they said was easy to follow. They felt reading the booklet pushed them to be more patient, which, in turn, they felt helped them be less aggressive, and not react angrily by shouting or threatening their children. ‘*Our children love to fight when playing games. The booklet showed me ways to resolve children’s fight successfully, and it worked*’. Several parents stated that the most useful information was on how to encourage good behaviour by praising their children for the behaviour that pleases them. These parents found themselves looking out for moments to praise their children, and, in turn, they began to appreciate and acknowledge such behaviour, and believed this further encouraged their children to act in a positive way. One parent described utilising this approach by saying, ‘*Positive words from us when our children fight has been very useful. Instead of the word “don’t”, it has been more useful using positive words and receiving positive attitudes and responses from my children*’.

Some of the parents mentioned that for them, the booklet did not necessarily present them with new ideas on how to parent, but rather refocused them on their important role, and reminded them how to take back responsibility and control of their family life. It reminded them of their key role as parents. One parent reflected on this with: ‘*The booklet reminded of the ways we should interact with our children when faced with challenges at home, as sometimes, parents forget and become too busy with other things. So, we are reminded of the effective ways to be*’. Furthermore, some parents described their home life as chaotic, as they were faced with many roles and responsibilities, leaving some to not even realise how much they needed such parenting tips to direct them on how to take control of their parenting, and how to better care for their children.

As for experiences with watching the video sketches, all interviewed parents described them as very useful, and that they helped make the tips provided in the detailed booklet more concrete. They stated that watching the actors roleplay the different scenarios of child and caregiver interactions brought the booklet tips to life, and helped them further understand how they can implement such interactions with their own families. Several of the parents also described how the videos further clarified some of the techniques described in the booklet so they further understood how to implement them. One parent said: ‘*Sometimes [in the booklet], we wonder “what do they mean by that?” but with videos, there are examples which made us feel like “this is what they mean”.*

Parents also described the video sketches as being useful in illustrating the importance of the choice of words to use, as well as the intonation and attitude in their interaction with their children. As one parent said: ‘*Watching the actors perform, seeing the language they used and the way they looked and touched their children brought the techniques to life*’. A key feature of the videos was that they first began with illustrating a negative roleplay of how one should not be interacting with their children, and then followed through with a positive roleplay demonstration. All parents noted that seeing the negative and positive roleplays helped them understand the differences between how to interact with their children better. One parent described how they felt it illustrated how by making simple small changes in the way they behave with their children might have big effects on children’s response to any challenging situation. This parent shared: ‘*The video showed me how just changing the way a parent’s voice responds to a child’s misbehaviour can immediately extinguish a very highly possible big argument or negative encounter. It is really the small things in our actions that matter*’.

The findings of this study are useful to further detail Indonesian parents’ experiences caring for children during a global pandemic. The parenting experience reflected in this study parallels other recent research on parenting during the COVID-19 pandemic [9], as well as previous research exploring the parenting experiences of caregivers during times of high-stress contexts, such as parenting through conflict and displacement contexts [10]. For example, caregivers reflected on changes in their children’s behaviours and emotions, and their frustrations at not knowing how to meet their children’s new needs. In addition, similar to previous research in refugee contexts, caregivers felt a sense of frustration and loss of control as they navigated their children’s new needs, and their own challenging emotions of sadness and anxiety [11,12]. Aggression features commonly in studies with families living in high-stress contexts [2], and this population was no exception, as parents described using more aggressive parenting during the lockdown period, but reflected on their ability to reduce such aggressive encounters after engaging in the parenting information. They went on to describe using a more compassionate and patient approach in their parenting, again often reflected in adult caregivers that engage in family skills information [13]. Furthermore, the study additionally availed an overview of perceptions towards a parenting information booklet and associated videos. Such findings avail further reassurance for the value and need of such material for families facing such circumstances.

Another important outcome of this study is that the booklet and video content are considered ‘light-touch’ parenting information when compared against more intensive multi-session parenting interventions. These results add a context of feasibility, and the ease of scale-up to the previously reflected elements of usefulness and value. Despite being ‘light-touch’, in that parents themselves were leading reading the booklet and watching the videos in their own time, and potentially only once, parents still described positive changes to their parenting, as well as positive responses by their children. This effect of even ‘light-touch’ information making a difference to the parenting experience and family interaction has been previously seen in a study utilising another ‘light-touch’ intervention consisting of a parenting booklet and conversation group seminar with families in conflict/post-conflict settings living in Palestine [14]. These results further parallel that of a recent study that sought to explore the usefulness of the ‘COVID-19 Playful Parenting Emergency Response initiative’, a multi-agency initiative intended to provide light-touch parenting information globally [15]. The results indicated positive changes in caregiving and child interactions, and the transformation of everyday interactions. That said, such as with any qualitative study, there is always the need for researcher awareness of the possibility that caregivers may have been influenced by social desirability bias to report positive changes. Despite this, the implications of this finding are exciting, and pave the way to the engagement and implication of more intensive parenting information interventions in high-stress contexts, in which we might expect that the effect they have will seemingly be even greater with even further exposure opportunities.

We enter 2022 with news of further variations of SARS-CoV-2, bringing with them the possibility of extended lockdowns around the globe, and with news of the need for everyone to adapt to a “new normal”. We collectively recognise that our journey navigating through this COVID-19 pandemic may remain a long journey ahead, and even then, this will certainly not be the last virus to threaten us [16]. Despite the challenges we face, there is promising news for families globally, as agencies and policy-makers begin to recognise, more and more, the importance of supporting families with the appropriate skills to navigate extreme stress contexts with effective strategies. By doing so, we reduce the likelihood of compromising child and adult mental health and wellbeing, and give every family the chance to thrive.

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
