# Peer review of "Caring for Your Child during COVID-19—Utilizing a Light-Touch Parenting Resource during Lockdown in Indonesia"

_ijerph, 2022, doi:10.3390/ijerph19074046_

Round 1
Reviewer 1 Report
This study describes a feasibility study on a booklet aiming to enhance parenting skills and build family harmony in the Covid-19 context.
Some more detail on procedures could be given.For instance, it is indicated that parents "were provided with both printed and electronic copies of the ‘Caring for your child dur- 105 ing COVID-19 lockdown’ booklet, as well as access to the associated video sketches", but not what specific instructions were given. Were parents instructed to read the booklet? In what way was the goal of the study explained? Hw were the researchers distributed over interviews: did they both interview half, or were they doing all interview with 2 researchers present?
I also miss some discussion of research reflexivity. To what extent can the (positive) evaluations provided by participating parents possibly have suffered from some social desirability bias?
The content of both the booklet and video sketches were not really described in detail. I think it would be helpful to add the booklet in an appendix and add a more detailed description of the video sketches.
Author Response
Reviewer 1:
This study describes a feasibility study on a booklet aiming to enhance parenting skills and build family harmony in the Covid-19 context.
We thank the reviewer for their time and useful comments.
Some more detail on procedures could be given. For instance, it is indicated that parents "were provided with both printed and electronic copies of the ‘Caring for your child during COVID-19 lockdown’ booklet, as well as access to the associated video sketches", but not what specific instructions were given. Were parents instructed to read the booklet? In what way was the goal of the study explained? How were the researchers distributed over interviews: did they both interview half, or were they doing all interview with 2 researchers present?
Thank you for this useful comment that has allowed us to elaborate further. We have added information on:
-participants received instructions to read the booklet first and followed with video a week after.
-Each researcher conducted 15 interviews
-Participants first received a participant information sheet
I also miss some discussion of research reflexivity. To what extent can the (positive) evaluations provided by participating parents possibly have suffered from some social desirability bias?
We have added:
‘That said, such as with any qualitative study, there is always the need for researcher awareness of the possibility that caregivers may have been influenced by social desirability bias to report positive changes. Despite this, the implications of this finding are exciting and pave the way to the engagement and implication of more intensive parenting information interventions in high stress contexts, in which we might expect that the effect they have will seemingly be even greater with even further exposure opportunities’.
The content of both the booklet and video sketches were not really described in detail. I think it would be helpful to add the booklet in an appendix and add a more detailed description of the video sketches.
-When mentioned in the article, the booklet has an associated reference that is a direct link to the booklet.
-It was useful to see feedback on the need for more information on the videos, thank you. We have now added ‘Nine short video sketches were developed in the local language (Bahasa) in Indonesia to highlight key topics in the booklet in an engaging and clear way and to overcome any challenges with low caregiver literacy. These topics included giving warmth and support, giving praise, spending time together, encouraging good behaviour, fighting and aggression, maintaining routines, encouraging play, fears and night disturbance and relaxation techniques. The duration of each video varied from 3 to 6 minutes’.

Reviewer 2 Report
Thank you for the opportunity to review this commentary, which describes the experience of 30 parents in Indonesia in using the UNODC ‘Caring for your child in response to the COVID-19 lockdown’ booklet. The manuscript is generally well-written and reports qualitative evidence describing parent challenges during COVID-19 and the potential usefulness of this light touch intervention. Below I provide some suggestions that may further enhance this commentary
1. The authors should carefully proofread the manuscript for grammar/spelling errors
2. pg 2, last paragraph- citations would help bolster the arguments made - for instance can the authors provide a citation to link economic stress to parenting behavior?
3. Is it possible to provide links to the booklet, for interested readers to access? In addition/alternatively could there be more information about the specific tips/topics included in the booklet and videos (perhaps a table listing topics covered)- this would help contextualize what the parents were exposed to, which would help when interpreting that it was helpful
4. To contextualize the findings, it would help to have more information about the families- such as some indicator of family ses, mean, SD, and range of number of children in home and parent age, % parents employed
5. Was there any information about level of use of the booklet and materials, to help understand the extent to which there was engagement? or the range of levels of engagement? Or is there an estimate of the time it would take to read the booklet and watch all videos- to provide an understanding of what is meant by "light touch"
6. I think this manuscript would be strengthened if the authors attended to any limitations/challenges in disseminating/utilizing these resources that researchers or interventionists might want to keep in mind. I assume the researchers learned a lot from this implementation.
Author Response
Reviewer 2: Thank you for the opportunity to review this commentary, which describes the experience of 30 parents in Indonesia in using the UNODC ‘Caring for your child in response to the COVID-19 lockdown’ booklet. The manuscript is generally well-written and reports qualitative evidence describing parent challenges during COVID-19 and the potential usefulness of this light touch intervention. Below I provide some suggestions that may further enhance this commentary
We thank the reviewer for their time and useful comments.
- The authors should carefully proofread the manuscript for grammar/spelling errors
We have undertaken a further full proofread.
- pg 2, last paragraph- citations would help bolster the arguments made - for instance can the authors provide a citation to link economic stress to parenting behavior?
-Great idea! Citation added now.
- Is it possible to provide links to the booklet, for interested readers to access? In addition/alternatively could there be more information about the specific tips/topics included in the booklet and videos (perhaps a table listing topics covered)- this would help contextualize what the parents were exposed to, which would help when interpreting that it was helpful
-When mentioned in the article, the booklet now has an associated reference that is a direct link to the booklet.
-It was useful to see feedback on the need for more information on the videos, thank you. We have now added ‘Nine short video sketches were developed in the local language (Bahasa) in Indonesia to highlight key topics in the booklet in an engaging and clear way and to overcome any challenges with low caregiver literacy. These topics included giving warmth and support, giving praise, spending time together, encouraging good behaviour, fighting and aggression, maintaining routines, encouraging play, fears and night disturbance and relaxation techniques. The duration of each video varied from 3 to 6 minutes’.
- To contextualize the findings, it would help to have more information about the families- such as some indicator of family ses, mean, SD, and range of number of children in home and parent age, % parents employed.
We agree that this information would have been interesting, but for this study this information was not collected.
- Was there any information about level of use of the booklet and materials, to help understand the extent to which there was engagement? or the range of levels of engagement? Or is there an estimate of the time it would take to read the booklet and watch all videos- to provide an understanding of what is meant by "light touch"
Thank you for this useful comment that has allowed us to elaborate further. We have added information on that participants received instructions to read the booklet first and followed with video a week after.
- I think this manuscript would be strengthened if the authors attended to any limitations/challenges in disseminating/utilizing these resources that researchers or interventionists might want to keep in mind. I assume the researchers learned a lot from this implementation.
This is another really useful comment, thank you. We have added the limitation on potential social desirability bias and the importance of keeping this in mind when reviewing the positive impact this light touch intervention has shown.
